# Predicting vasovagal reactions to needles from video data using 2D-CNN with GRU and LSTM

**Judita Rudokaite**[1,2]*, **Sharon Ong**[1], **Itir Onal Ertugrul**[3], **Mart P. Janssen**[2], **Elisabeth Huis in 't Veld**[1,2]

**1** Department of Cognitive Science and Artificial Intelligence, Tilburg University, Tilburg, The Netherlands, **2** Donor Medicine Research, Sanquin Research, Amsterdam, The Netherlands, **3** Department of Information and Computing Sciences, Utrecht University, Utrecht, The Netherlands

\* j.rudokaite@tilburguniversity.edu

**Data Availability Statement:** Data cannot be shared publicly because this is a sensitive information containing each donor's face that

## Abstract

When undergoing or about to undergo a needle-related procedure, most people are not aware of the adverse emotional and physical reactions (so-called vasovagal reactions; VVR), that might occur. Thus, rather than relying on self-report measurements, we investigate whether we can predict VVR levels from the video sequence containing facial information measured during the blood donation. We filmed 287 blood donors throughout the blood donation procedure where we obtained 1945 videos for data analysis. We compared 5 different sequences of videos—45, 30, 20, 10 and 5 seconds to test the shortest video duration required to predict VVR levels. We used 2D-CNN with LSTM and GRU to predict continuous VVR scores and to classify discrete (low and high) VVR values obtained during the blood donation. The results showed that during the classification task, the highest achieved F1 score on high VVR class was 0.74 with a precision of 0.93, recall of 0.61, PR-AUC of 0.86 and an MCC score of 0.61 using a pre-trained ResNet152 model with LSTM on 25 frames and during the regression task the lowest root mean square error achieved was 2.56 using GRU on 50 frames. This study demonstrates that it is possible to predict vasovagal responses during a blood donation using facial features, which supports the further development of interventions to prevent VVR.

## Introduction

On average one in three adults is scared of needles [1] which unfortunately also makes them vulnerable to experiencing adverse emotional and physical responses during a needle-related procedure. These so-called vasovagal reactions (VVR) consist of nausea, dizziness, heart palpitations, hyperventilation, or even fainting with a loss of consciousness. VVR can be at least partly explained by sympathetic autonomic nervous system activations which can occur during stressful events such as an injection or blood draw. These reactions can result in physiological changes such as increased heart rate [2,3], sweating, nausea, pupillary dilation, changes in facial pallor or hyperventilation [4–6]. In contrast to other types of fearful stimuli or situations,

would breach patient's privacy if made publicly available. Request more information by contacting the data manager of the Donor Medicine Research department at Sanquin, the corresponding author Judita Rudokaite via email j.rudokaite@tilburguniversity.edu or dr. Elisabeth Huis in 't Veld via email e.m.j.huisintveld@tilburguniversity.edu. The dataset can be shared with researchers who meet the criteria for access to confidential data. Partial anonymized preprocessed dataset can be found at https://dataverse.nl/dataset.xhtml?persistentId=doi:10.34894/WXPXKW.

**Funding:** This research was funded by the ZonMW Veni project "FAINT" (project reference: 016.186.020) and Stichting Sanquin Bloedvoorziening (grant PPOC19-12/L2409). Funders did not have any role in study design, data collection and analysis, decision to publish, or preparation of the manuscript.

**Competing interests:** Authors Dr. E.M.J. Huis in 't Veld and J. Rudokaite are founders of AINAR B.V. The other authors declare no competing interests. we would like to declare that this commercial affiliation does not alter our adherence to PLOS ONE policies on sharing data and materials.

during needle-related procedures specifically physiological changes can also occur which are more likely to be due to increased *para*sympathetic activity, such as drops in heart rate or blood pressure [3]. During a vasovagal reaction, the patient or the blood donor experiences a sudden drop in arterial blood pressure and as a result, it reduces blood flow to the brain and the person reports feeling lightheaded or dizzy [7]. A loss of consciousness in such a case is called a vasovagal syncope. Even though vasovagal reactions and syncopes are usually benign in nature, they can lead to more serious complications for both patients and healthcare providers, including head or fall-related injuries in the short term [8] but also the refusal, avoidance or aborted medical procedures in the future [9,10].

## Risk factors of experiencing VVR

One of the main risk factors of experiencing vasovagal reactions is needle fear [9,10]. Needle fear may be triggered by a combination of environmental and personal factors such as genetic predisposition [11,12], frequent exposure to needles or blood draws [13], or personality characteristics such as heightened sense of anxiety [14]. Other psychological states such as anticipated anxiety, fear of blood and injury, fear of blood draws, perceived blood loss, pain, anticipated pain, and anticipated disgust, have also been associated with increased risk of experiencing vasovagal reactions [15–18] as well as demographic characteristics such as (female) gender, younger age, and lower BMI [9,10,19,20].

To summarize, Thijsen & Masser (2019) [10] divided the risk factors into three categories such as 1) donor characteristics that are observable like gender or ethnicity, 2) donor characteristics that may not be immediately known such as fear of needles and 3) contextual features of the procedure itself such as increased waiting time or how experienced medical staff were in helping people with needle fear. In terms of prevention, unfortunately, many of these factors cannot be targeted for change and can at most provide a guide to who would benefit from an intervention most. The main targets for current interventions, however, are behavioral risk factors such as lack of sleep prior to the needle-related procedure, food and water intake and caffeine consumption [21,22].

## Current intervention strategies to prevent vasovagal reactions

Currently, well-known interventions aimed at preventing VVR are mostly geared towards donors, who may faint due to the loss of 500ml blood, and these techniques include water loading or actively applied muscle tension, which are meant to combat the symptoms related to the loss of blood pressure that may occur as a result. Water loading refers to a technique where a person is asked to ingest around 500 ml of water within 30 minutes or less prior to the procedure [23,24]. Applied muscle tension consists of repeated contractions of muscles in the legs and/or abdomen in order to increase blood pressure [23,25]. Even though research shows that these techniques could work for a subset of donors [25], a meta-analysis suggested that these techniques are insufficient for the majority of donors [23] and do not reduce the rate of syncopic reactions [25].

Other preventive strategies provided by healthcare professionals range from providing extra information, social support or distractions or even administering calming medication such as low doses of benzodiazepines that allow the patients to reduce their anxiety levels [26]. Although benzodiazepines reduce the number of vasovagal reactions by addressing underlying fear and anxiety, some side effects could make them less favorable as a prevention strategy, especially, as blood draws or immunizations are quick procedures. All previously applied preventive techniques might be effective, however, they are costly in terms of extra time required by the staff. Hence, there is an enormous demand for new prevention methods.

## Anticipatory physiological response as a new VVR prevention strategy

As risk factors are difficult to affect it may be more promising to look at *when* it is best to intervene. Research shows that anticipatory fear, anxiety and stress and a history of previous vasovagal reactions are one of the most important risk factors for experiencing vasovagal reactions [10,14–16,27]. This is corroborated by studies from Hoogerwerf et al. (2018; 2017) [28,29] who assessed psychological, hormonal and psychophysiological stress markers in donors throughout a blood donation and found that the objective stress markers already occur at a very early stage in anticipation of the needle insertion, at which time they peak. For example, the levels of systolic blood pressure and cortisol levels increased towards needle insertion and then decreased after the blood donation [28,29]. In addition, higher systolic blood pressure and pulse rate were found in women and first-time donors, who are at the higher risk of experiencing VVR [9,10,19,20].

To make matters even more challenging, these physical reactions that are targets for change, such as heart rate, heart rate variability, respiratory signal, skin temperature, or brain waves measured with EMG or EEG [30] are automatic, difficult to self-report [7,31] and require devices such as heart-rate monitors with attached electrodes on the patient, EEG caps, or respiratory vests. In one of our previous studies, we mimicked a blood donation using an experimental 'virtual' rubber arm illusion, capturing the participant with an infrared thermal imaging camera. This experiment showed that changes in facial temperature could serve as early indicators for vasovagal reactions [32,33]. Specifically, facial temperature fluctuations in the area under the nose, chin and forehead are associated with increased risk of experiencing VVR [33]. Although thermal cameras can be used in various light conditions including low-light or even complete-darkness and are less likely to be affected by any changes in person's appearance, visible light imaging provide a much more detailed information about the visual appearance, including any skin changes, have higher resolution, and, more importantly, are much cheaper and more widely used than thermal imaging, making them a preferred option for an intervention.

There is a lack of user-friendly solutions able to monitor to what extent patients are (starting to) experience early signs and symptoms of VVR in real time. This is especially important as anticipatory processes take place when patients are 'out of sight' of medical staff, for example when they are in the waiting room. Given that the procedure at blood collection centers usually lasts around 10–15 minutes in total, which is long compared to the time it takes to collect a sample for a medical test, it would not be feasible and practical to implement a solution that requires more preparation time than the procedure itself.

## Applying deep learning methods for automated video analysis for a biofeedback-based serious game intervention

The most desirable solution would be using a non-invasive and low-cost method to monitor donors or patients. In order to address the lack of interventions that address the anticipatory risk factors for VVR in practice we developed a solution for people with needle fear that is able to not only identify covert symptoms of emotional and physical reactions at a very early stage, but also to immediately give them a tool which can help them to prevent the escalation into a vasovagal event. To achieve that, we aimed to implement the best performing model in a serious game for smartphones, which through the facial video input from the front-facing camera, controls a biofeedback mechanism which will help the player get control over their impeding VVR in an early stage.

Video recordings of the face contain many types of useful information. For example, it contains information about facial expressions which, even when they are very subtle, enable the detection of anxiety, stress, fear, pain, and vasovagal reactions [34–39]. In addition, the face contains information of head movements, eye-gaze direction, changes in facial colors such as paleness, etc. With the recent developments in the field of deep learning and in particular automatic face analysis [40–42], it has been shown possible to predict not only mental health conditions such as depression, anxiety, or obsessive-compulsive disorder [43], but also physical symptoms such as pain [44–46]. These models can potentially be used as valuable tools for clinical diagnosis and for monitoring and altering physical responses in real time [32]. Specifically, we found a significant association between vasovagal reactions and changes in facial temperature [33] as well as facial micro-expressions [34]. Both changes in thermal fluctuations and facial action units recorded prior to blood donation showed promising results in predicting vasovagal reactions that occur during or after blood donation [32,33,41].

However, assessing the risk of VVR is only one part of the solution. Even better would be if the patient can use this information to prevent the VVR from happening. This can be achieved through biofeedback. Biofeedback is a self-regulation technique that allows individuals to gain control over their typically involuntary physiological responses by providing real-time feedback on their neurological or physical processes [47]. The main goal of biofeedback is to help individuals reduce the arousal of their sympathetic nervous system—the system responsible for the "fight or flight" response—so that they can learn to consciously regulate processes like heart rate, muscle tension, or blood pressure [48]. The feedback consists of a visual (or auditory) reflection or representation of the person's neurological or physiological state or processes (e.g. heart rate, breathing patterns, muscle tension, skin conductance, electroencephalogram, or skin temperature, just to name a few) measured with sensors or devices. By seeing or hearing these physiological signals in real time, patients can experiment with strategies to manage their stress or anxiety and bring their physiological state back into balance. For example, biofeedback has been successfully used to treat conditions like stress, anxiety [49], substance abuse [50], seizures, epilepsy [51], and ADHD [52]. Biofeedback can be delivered via computer screens or mobile apps and is often incorporated into interactive environments such as video games [53]. In these games, the stimuli adjust based on the player's bodily responses, allowing the individual to practice managing their physiological reactions in a controlled and engaging environment.

Our serious game solution for needle fear (called AINAR, Artificial Intelligence for Needle Induced Fainting) continuously assesses the likelihood of experiencing a vasovagal reaction (VVR) through the model, which gets its input from the front-facing camera, which is then reflected in the weather, which can be sunny, rainy, or snowy. The player's task is to keep the weather nice and sunny. If it starts to rain, the player can experiment with different relaxation techniques or cognitive strategies to transition from a state of fear or stress to calmness, thus learning to control their body's automatic responses. The aim is to provide the feedback as often as possible with little delay, therefore, we aim to investigate what is the shortest length of video that is required for acceptable VVR prediction. This approach allows for real-time learning and adaptation, making biofeedback a powerful tool for overcoming anxiety and stress-related conditions.

## Use of machine learning when donors with high VVR scores are rare

The prevalence of severe VVRs in blood donors ranges from 0.1% to 0.5% [54]. This poses limitations on training machine learning models from scratch given that deep learning models are data-hungry. To train a model from scratch, a large amount of training data of donors with

both high and low VVR scores is required, which is difficult to obtain. To overcome this limitation, transfer learning will be applied. Transfer learning is one of the machine learning techniques where a model, initially trained on one task, is used for a different task with some additional tuning. The main advantages of transfer learning are that it requires smaller datasets and shorter computational time, and may improve the overall performance since the initial model was trained on a much larger dataset [55]. There are various transfer learning models such as ResNet50, VGG16, VGG19, Inception, or Xception. All pre-trained models are usually based on a Convolutional Neural Network (CNN) architecture, which is not only one of the most popular neural network models used in solving image classification problems, but it also requires fewer parameters and shorter training time than other conventional neural networks. Instead of using a fully connected network of weights from each pixel, a CNN scans a small patch of the image which is used to scan the entire image. We selected ResNet152 and Xception models because they have deeper architectures than ResNet50, VGG or Inception models and can capture more complex features and patterns in data [56,57], potentially leading to better performance in capturing subtle facial changes. In addition, ResNet152 and Xception are trained on larger datasets with more diverse images, which allows them to learn richer and more generalizable representations compared to shallower models like ResNet50 and VGG [56–58].

Two-dimensional convolutional neural networks (2D-CNN) architectures are used for extracting spatial features whereas the recurrent neural network (RNN) is often used for capturing temporal features [59]. However, the issue with applying RNN directly on long sequences is that the gradients are propagated over so many stages that they might vanish or explode [60–62]. Vanishing gradients make it difficult to see in which direction the parameters should move to improve the loss functions and exploding gradients make the learning unsteady [56]. To mitigate this problem, Long Short-Term Memory (LSTM) or gated Recurrent Unit (GRU) models are often used [62]. The GRU uses fewer parameters than the LSTM and therefore has faster training time, but the LSTM is a more complex model that could capture prominent features more accurately [59]. Thus, the combination of a pre-trained 2D-CNN model with either GRU or LSTM is often used in similar studies [63]. Therefore, in our study, we aim to use pre-trained deep learning models to extract facial features from the video recording and use these spatial features for training LSTM and GRU models to be able to predict low or high VVR levels.

To conclude, in this study, we aim to assess to what extent video classification algorithms such as 2D-CNN pre-trained models with GRU and LSTM can be used to correctly classify the level of VVR a donor is experiencing. Additionally, since the long-term goal of predicting VVR levels would be to implement them into a biofeedback-based solution able to continuously monitor the player and to give real-time visual feedback, the duration of the video recording plays a crucial role. The shorter the duration of the video required, the more biofeedback signals can be sent to the user. Therefore, we evaluate the shortest duration necessary for the model to achieve the highest performance.

## Methods and materials

### Participants

Participants were recruited from the regular blood donor pool from Sanquin, the not-for-profit organization responsible for the blood supply in the Netherlands. The study took place at four blood collection centers (BCC Leiden, 's-Hertogenbosch, Zwolle, and Utrecht). Donors who adhered to the following inclusion criteria for one of three predefined subgroups were invited to participate: (1) the control group (N = 85); consisting of donors who donated

between 5 and 10 times, and who never experienced vasovagal reactions in the past, (2) the sensitive group (N = 65); donors with 5 to 10 previous donations who experienced a VVR at their previous donation, and (3) the new donor group (N = 137); consisting of first-time donors.

## Ethics approval

The study was approved by the Ethics Advisory Board of Sanquin and the Research Ethics and Data Management Committe (REDC#2019172) of the School of Humanities and Digital Sciences at Tilburg University. This study was performed in line with the principles of the Declaration of Helsinki and written informed consent was obtained from all participants. The data collection started on 11[th] September 2019 and lasted until 30[th] November 2022.

## Procedure

Interested donors contacted the data manager for an appointment and received information about the study. On arrival, participants signed the informed consent and then completed a questionnaire containing items regarding needle fear and several personality questionnaires. Next, the donors proceeded with the regular blood donation procedure consisting of several phases: registration, health check at the donor physician, blood donation, and cafeteria visit. This resulted in seven distinct stages during which video and VVR were recorded (see Fig 1). Specifically, at each stage donors were recorded using a regular video camera and had to verbally self-report their VVR score (see section Materials and measures, Vasovagal reactions for more information on the VVR score). During stages 1–3 and 7, the video recordings lasted around 1 to 2 minutes. In stages 4 to 6, the donors were seated in the donation chair, where the video recording was continuous and lasted between 5 and 27 minutes (or, rather the entire blood donation procedure). During this long recording, the VVR levels were accessed 3 times: at the needle insertion (stage 4), around the extraction of 300 ml blood (stage 5), and during needle uncoupling (stage 6). Throughout the entire procedure, donors were free to behave as they normally would. Verbal VVR ratings were noted by the data manager and recorded using a smartphone voice recorder.

## Materials and measures

**Vasovagal reactions (VVR levels; based on the Blood Donation Reactions Inventory (BDRI); [31]).** At each stage (see Fig 1) participants were asked to rate 8 questions regarding experienced physiological reactions (faintness, dizziness, weakness, lightheadedness) and emotional reactions (fear, stress, tension, and nervousness), on the Likert scale from 1 (not at all) to 5 (extremely), resulting in a score between 8 and 40 per time point.

**Video recording.** The videos were recorded at 25 frames per second using the Nikon Coolpix AW130. The camera was installed on a tripod at a distance of about 1m from the donor. Donors were free to behave as they normally would throughout the whole procedure.

## Video data preprocessing

To train the models to predict vasovagal reactions, video recordings were semi-automatically preprocessed to separate them into seven distinctive stages that served as input for the model. As the lengths of the videos varied, all videos were shortened to 45 seconds. The last 45 seconds were extracted from the original videos at stages one, two, three and seven since those were separate short recordings and in the first few seconds the donors were often positioning themselves in their chairs. The starting points for stages four, five and six were selected manually

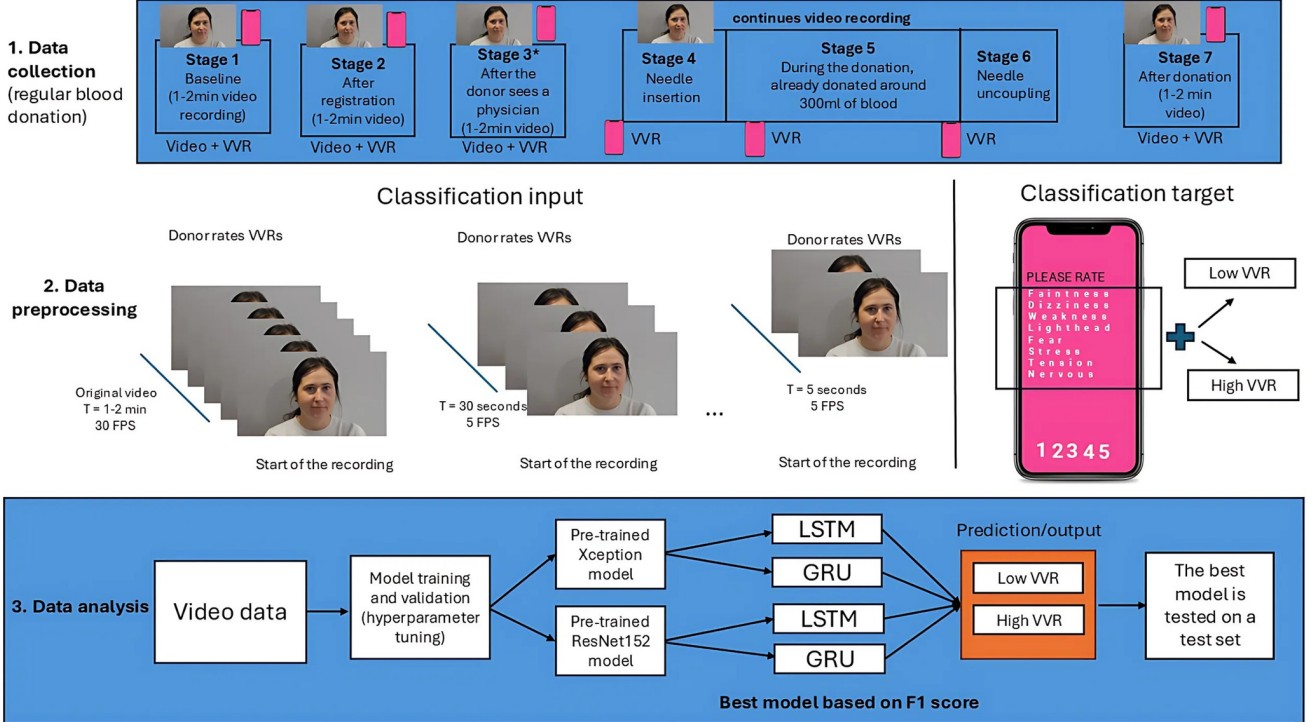

**Fig 1. An overview of the data collection and used methodology for data analysis. Stage 1:** Data collection shows an overview of the blood donation testing procedure and stages. At each of the seven stages, donors reported their VVR levels, and a video recording was made. At each stage, the video recording lasted 1 to 2 minutes. From stage 4 until stage 6, the recording was continuous, lasting between 5 and 27 minutes. *The procedure is slightly different per BCC. At two locations, donors were brought directly to the donation chair after the physician check and hence no recording at stage 3 was made. If donors had to return to the waiting area after the physician check, they would be asked for an additional self-reported VVR score. An additional recording of 1 to 2 minutes would be made. Stage 2: Data preprocessing shows an overview of the video preprocessing steps. At each step, the earliest frames were removed. i.e. the frames closer to self-report VVR measures are kept for the analysis. In addition, the number of frames was reduced from 30 FPS to 5 FPS. Stage 3: Data analysis shows an overview of model training and evaluation steps.

from the original continuous recording and the first 45 seconds of each stage were extracted, specifically, 45 seconds before the needle insertion (stage 4), 45 seconds around 300 ml of donated blood (stage 5), and 45 seconds around needle uncoupling moment (stage 6). We eliminated redundant frames by reducing the frame rate to 5 frames per second for each of the 45 second recordings. This resulted in 225 frames per video with a resolution of 1920 x 1080 pixels. In total, 1945 videos were used for further analysis.

We tested the classification and regression performance on different video lengths: 45 seconds (N = 225 frames), 30 seconds (N = 150 frames), 20 seconds (N = 100 frames), 10 seconds (N = 50 frames), and 5 seconds (N = 25 frames). The last fraction of each video recording (i.e., the fraction that was closest to self-reported VVR) was selected to shorten the videos. For instance, 45 seconds before needle insertion, 30 seconds before needle insertion, 20 seconds before needle insertion, etc. (see Fig 1).

## Deep learning approach

**Transfer learning using Xception and ResNet152.** In our study we used two pre-trained models based on Convolutional Neural Network (CNN) architecture (Xception and ResNet152) for spatial feature extraction. Xception is a convolutional neural network architecture that is an extension of the Inception architecture. It is trained on the ImageNet database

that is 71 layers deep and relies solely on depth wise separable convolution layers [56]. Residual Network (ResNet) is a feedforward network which consists of residual blocks with direct connection that skips some layers in between and, in this way, generates new inputs and outputs [57]. The premise of this network is to produce better accuracy without increasing the complexity of the model with ResNet152 achieving the best performance among other ResNet architectures [57].

These two pre-trained models were selected because they tend to perform better in comparison to VGG and other Inception models [57,58].

**Feature extraction using 2D-CNN with GRU and LSTM models.** Each video frame was passed to a pre-trained model after it was resized to fit the default size of the models, specifically 299x299 for Xception [64] and 224x224 for ResNet152 [57]. Both pre-trained models returned vectors containing extracted features of size 2048, which were then used to train LSTM and GRU models:

1. LSTM stands for long short-term memory and is a type of recurrent neural network (RNN) that is capable of learning order dependence and avoiding long-term dependency problems by having self-connected hidden layers containing memory cells and corresponding gate units [65].

2. GRU stands for Gated Recurrent Unit, which is an advancement of the standard recurrent neural network (RNN) that uses the reset and the update gates to overcome the issue of vanishing and exploding gradients [66].

The architecture of both the GRU and LSTM consisted of two GRU or LSTM layers and a dropout layer as it previously yielded best results [67]. The optimizer, activation functions and loss functions were predefined, but different for classification and regression models.

1. Adam was chosen as the optimizer for the classification task, with a learning rate of 0.0001, as this is computationally efficient and suitable for a model with many parameters [68]. The selected activation function was a sigmoid that produces a number between zero and one, where any value below 0.5 is classified as negative and above as positive. The binary cross entropy was specified as a loss function where the target of predictions is zero or one and uses the sigmoid as the activation function for making these predictions [69].

2. For the regression task, Root Mean Squared Propagation (RMSprop) was chosen as the optimizer, which is a gradient-based optimization technique that uses an adaptive learning rate. The selected activation function was Relu with root mean squared error as a loss function. RMSE is a square root of the difference between the true dependent variable and the predicted dependent variable.

The number of units used, learning rate, dropout rate and epochs were determined empirically. We have tested the following hyperparameters: learning rate of 0.0001, 0.001 and 0.01, dropout rate of 0.1, 0.3, 0.5, batch size of 32 and 64, and epochs of 50, 100, and 200. For both GRU and LSTM models the selected number of units was 32 and 16, the dropout rate was 0.1, learning rate was 0.0001, batch size was 32, and number of epochs was 100.

**Model training, validation, and evaluation.** The dataset was split into a training (80%) and test (20%) set, on which the model performance was assessed. We used validation splitting to automatically reserve the fraction of the training data for evaluating the loss and model metrics at the end of each epoch. We selected 20% of the data for testing by taking the last 20% of samples of the arrays received by the model, before any shuffling.

The original dataset contained 1945 videos of which 592 (30%) belonged to a high VVR class sample. The data was split based on subject identification number to ensure that the same

participants would not appear in both training and testing sets. Due to class imbalance, we applied video data augmentation on the high VVR cases in the training data. Specifically, we generated new videos by applying horizontal flips and adding some noise (Video Augmentation Library; [70]). Thus, after data augmentation, we split our sample again and then trained our model on 1742 low VVR class and 1491 high VVR class examples. The test set was not manipulated in any way and contained 158 low VVR class and 145 high VVR class examples.

To evaluate model performance and account for class imbalance, we used the following metrics for evaluating the **classification task**:

1. Precision—the proportion of correctly predicted high VVR scores of all high VVR scores.

2. Recall—the proportion of correctly identified high VVR donors out of all donors classified as high VVR donors.

3. F1 score, which is the harmonic mean of precision and recall.

4. AUC-PR score, which is the Area Under the Precision-Recall Curve that summarizes a precision-recall curve as the weighted mean of precisions over all recall values. The higher the AUC-PR score, the better the overall performance of the model with 1.0 being a perfect model.

5. Matthew's correlation coefficient (MCC), which is a contingency matrix method of calculating Pearson correlation coefficient between actual and predicted values. This measure provides a high score only if the binary predictor is able to correctly predict the majority of instances of both low and high VVR groups [71,72]. This metric ranges from 1- to +1 where -1 indicates total disagreement between predicted and actual values, 0—predictions that are no better than a random selection, and ±1—the perfect agreement between predicted and actual values.

We reported precision, recall, F1, AUC-PR and MCC scores on the test sets at each tested time interval.

To evaluate which parts of the image are important for classifying low and high VVR groups, we occluded some regions in the image and re-evaluated model performance as suggested in Ertugrul et al., (2020) [73]: the more model performance drops after application of the occlusion, the more important the region is for the classification. We applied larger rectangles (100x100) around the face (containing only background information) and smaller rectangles (80x80 and 60x60) within the face region. An overview of the occluded regions is shown in Fig 2.

The Root Mean Squared Error (RMSE) was used to evaluate the performance of the regression task. The RMSE is considered the standard error metric for numerical predictions. Note that the RMSE was calculated in the unit of measurement of our outcome of interest (VVR score), and therefore this measure is directly interpretable. The lower the RMSE values, the better the performance of the model.

**Statistical analysis.** For statistical analysis the data normal distribution was verified applying the Shapiro-Wilk test, rejecting the null hypothesis (normal distribution) at the 5% significance level. An ANOVA test was used to determine if there was a statistically significant difference between donor groups (categorical variable) by testing for differences of means using a variance. Questionnaire data were analyzed using RStudio (2020) [74].

The Friedman test also known as the non-parametric repeated measures ANOVA was used to compare the performance of the classifiers and evaluate whether there are significant differences in the performance.

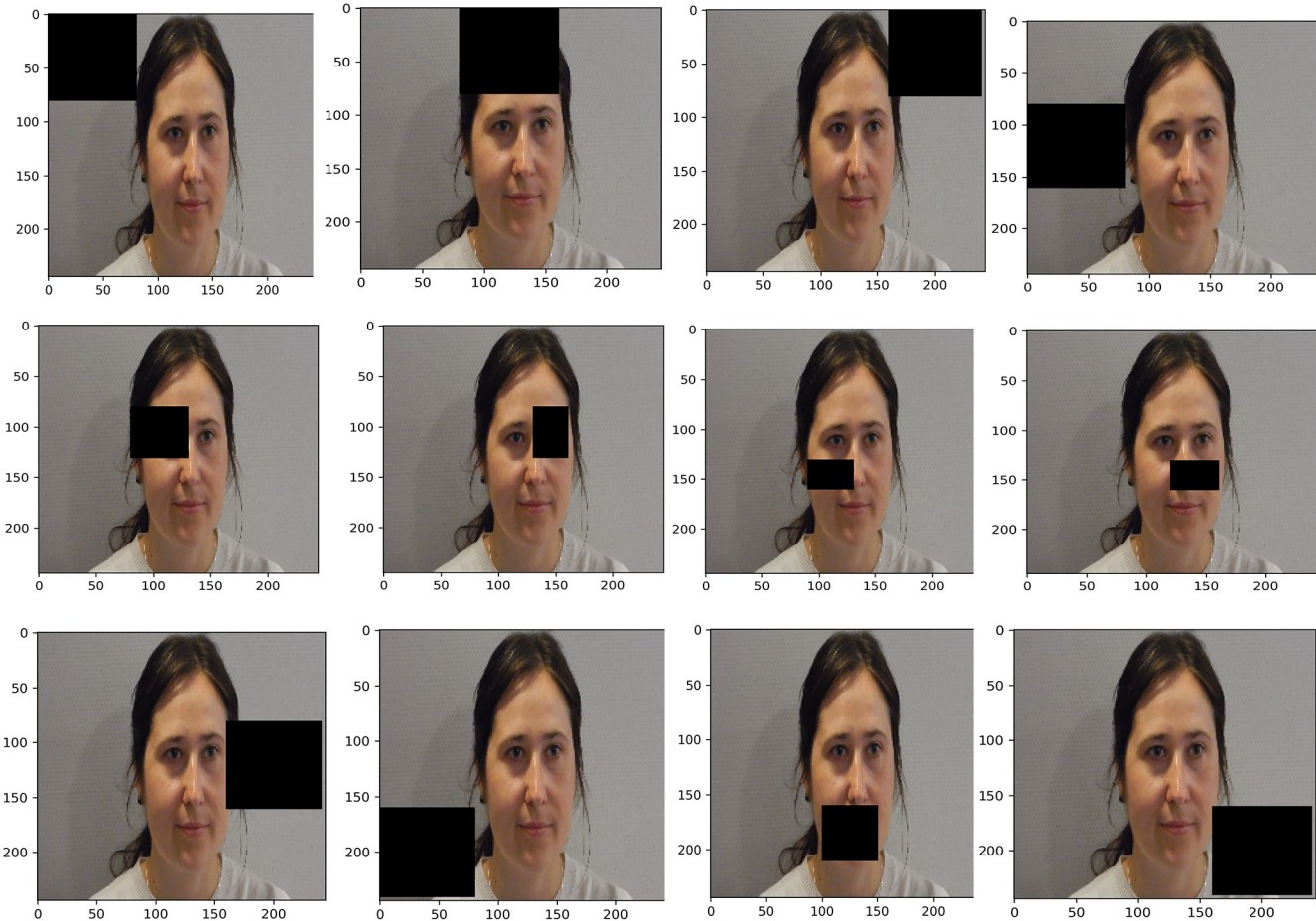

**Fig 2. Overview of the regions occluded for the assessment their importance for classification.** After alignment, the pixel values in the black boxes shown were set to zero.

## Results

### Participants

The data was collected from 287 blood donors in total (control group: n = 85 sensitive group: n = 65, new donors: n = 137). No statistically significant differences in gender ($F_{(2)}$ = 2.76, p = .065), blood collection centers ($F_{(2)}$ = 1.57, p = .21), or age ($F_{(2)}$ = 1.66, p = 0.19; M = 38.98, SD = 13.45) were found between the groups.

### VVR levels

VVR scores were positively skewed, reflecting a high proportion of blood donors who reported low VVR scores (M = 11.61, SD = 3.81, median = 11.0; min = 8, max = 40, see Fig 3). The raw VVR scores were directly used for regression analysis. For the classification task, the videos were split into representing low VVR (N videos = 1900, VVR score < = 11) and high VVR (N videos = 1636, VVR level > 11) groups.

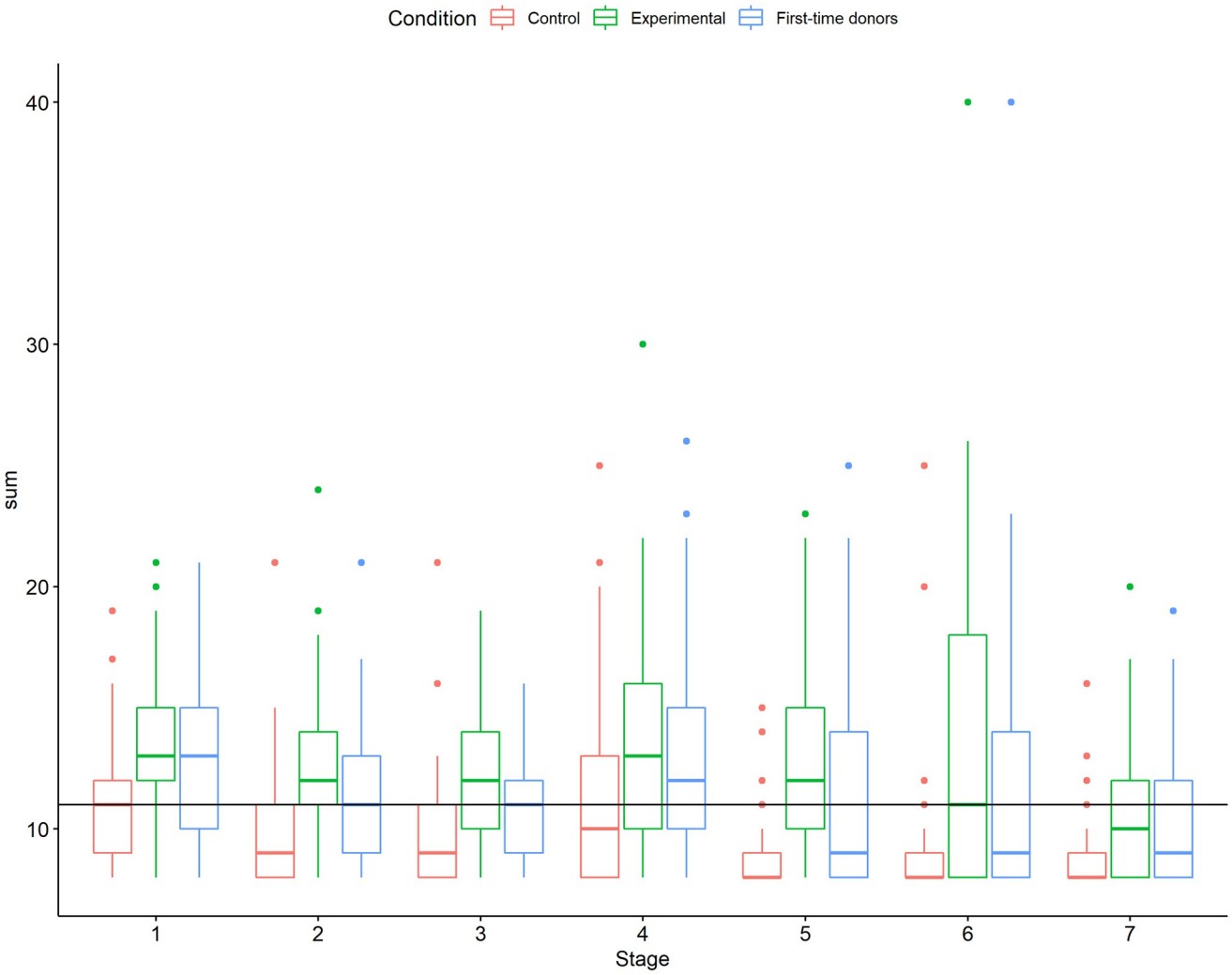

**Fig 3. Distribution of VVR scores per stage.** Distribution of VVR ratings per stage and group. The dots above the box represent the outliers per group. The black line represents the cut-off on which the low vs high VVR groups were split.

## VVR classification results

We applied two pre-trained 2D CNN models with GRU and LSTM on various lengths of image sequences in order to classify low and high VVR groups. The overall results of the classification are presented in Table 1.

The classification report on the test set showed that the models performed slightly better in classifying low VVR classes. However, the main target group in this study is the high VVR class because we aim to identify people who are at risk of experiencing VVR symptoms. Therefore, we evaluated the model performance on the high VVR class. Since the performance was similar across all video durations, we also evaluated the changes in precision and recall on the shortest duration (see Fig 4). We conducted the Friedman test to determine if there were any statistically significant differences in performance metrics (such as F1 score, precision, and recall) across four models. We selected the scores obtained using the shortest video duration on classifying the high VVR class. We found no statistically significant differences in the performance of the models (Friedman test statistic = 5.64, p = 0.6).

**Table 1. The 2D-CNN performance on the test set classifying low (n = 158) vs high (n = 145) VVR classes using pre-trained Xception and ResNet152 models with GRU and LSTM on various video sequences ranging from 225 to 25 frames.**

| Model | Number of frames | Group | Precision | Recall | F1 | AUC-PR | MCC |
|---|---|---|---|---|---|---|---|
| Pre-trained Xception model with GRU | N = 225 | High VVR | 0.76 | 0.64 | 0.70 | 0.79 | 0.47 |
| | | Low VVR | 0.73 | 0.85 | 0.79 | 0.84 | |
| | N = 150 | High VVR | 0.74 | 0.68 | 0.71 | 0.79 | 0.47 |
| | | Low VVR | 0.73 | 0.78 | 0.76 | 0.82 | |
| | N = 100 | High VVR | 0.79 | 0.61 | 0.69 | 0.79 | 0.48 |
| | | Low VVR | 0.71 | 0.85 | 0.77 | 0.83 | |
| | N = 50 | High VVR | 0.73 | 0.68 | 0.71 | 0.78 | 0.46 |
| | | Low VVR | 0.73 | 0.77 | 0.75 | 0.82 | |
| | N = 25 | High VVR | 0.70 | 0.70 | 0.70 | 0.77 | 0.43 |
| | | Low VVR | 0.73 | 0.73 | 0.73 | 0.85 | |
| Pre-trained Xception model with LSTM | N = 225 | High VVR | 0.83 | 0.63 | 0.71 | 0.82 | 0.53 |
| | | Low VVR | 0.72 | 0.81 | 0.76 | 0.82 | |
| | N = 150 | High VVR | 0.71 | 0.71 | 0.71 | 0.78 | 0.44 |
| | | Low VVR | 0.73 | 0.73 | 0.73 | 0.84 | |
| | N = 100 | High VVR | 0.74 | 0.63 | 0.68 | 0.77 | 0.43 |
| | | Low VVR | 0.70 | 0.79 | 0.74 | 0.83 | |
| | N = 50 | High VVR | 0.75 | 0.61 | 0.67 | 0.77 | 0.43 |
| | | Low VVR | 0.69 | 0.82 | 0.75 | 0.85 | |
| | N = 25 | High VVR | 0.73 | 0.64 | 0.68 | 0.77 | 0.43 |
| | | Low VVR | 0.70 | 0.78 | 0.74 | 0.83 | |
| Pre-trained ResNet152 model with GRU | N = 225 | High VVR | 0.71 | 0.70 | 0.71 | 0.72 | 0.31 |
| | | Low VVR | 0.73 | 0.73 | 0.73 | 0.80 | |
| | N = 150 | High VVR | 0.80 | 0.64 | 0.71 | 0.81 | 0.51 |
| | | Low VVR | 0.77 | 0.65 | 0.70 | 0.81 | |
| | N = 100 | High VVR | 0.76 | 0.66 | 0.71 | 0.79 | 0.47 |
| | | Low VVR | 0.72 | 0.80 | 0.76 | 0.79 | |
| | N = 50 | High VVR | 0.72 | 0.68 | 0.70 | 0.76 | 0.44 |
| | | Low VVR | 0.72 | 0.75 | 0.74 | 0.81 | |
| | N = 25 | High VVR | 0.79 | 0.66 | 0.72 | 0.81 | 0.51 |
| | | Low VVR | 0.71 | 0.74 | 0.73 | 0.79 | |
| Pre-trained ResNet152 model with LSTM | N = 225 | High VVR | 0.72 | 0.68 | 0.70 | 0.76 | 0.39 |
| | | Low VVR | 0.72 | 0.75 | 0.73 | 0.80 | |
| | N = 150 | High VVR | 0.88 | 0.63 | 0.73 | 0.84 | 0.58 |
| | | Low VVR | 0.73 | 0.92 | 0.82 | 0.81 | |
| | N = 100 | High VVR | 0.75 | 0.63 | 0.69 | 0.78 | 0.45 |
| | | Low VVR | 0.71 | 0.81 | 0.76 | 0.82 | |
| | N = 50 | High VVR | 0.81 | 0.62 | 0.70 | 0.81 | 0.51 |
| | | Low VVR | 0.71 | 0.87 | 0.78 | 0.79 | |
| | **N = 25** | **High VVR** | **0.93** | **0.61** | **0.74** | **0.86** | **0.61** |
| | | **Low VVR** | **0.73** | **0.96** | **0.83** | **0.80** | |

This shows that both pre-trained Xception and ResNet152 models overall performed similarly across all tested sequences with precision scores ranging from 0.71 to 0.93 and recall scores ranging from 0.61 to 0.71 for the high VVR class. The performance using GRU and LSTM was also similar with the highest F1 score of 0.74 and PR-AUC of 0.86 using a pre-trained ResNet152 model with LSTM on 25 frame sequence for high VVR class. ResNet152

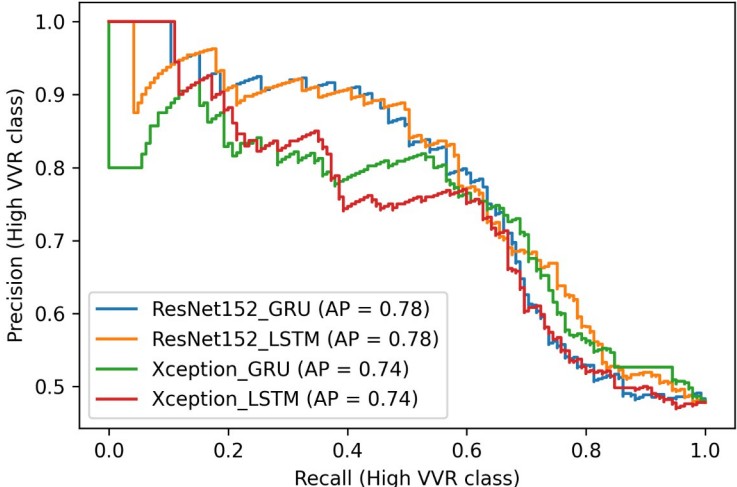

**Fig 4. The figure shows the precision-recall curves of all 4 models using the sequence of 25 frames.**

with LSTM showed high precision in identifying people who are at risk of experiencing VVR symptoms, however, it had a slightly lower recall score in comparison to other models. The results indicate that this model captures slightly fewer high-risk people than other models, however, it is very accurate in identifying those who are at risk. On the other hand, this model performed well on the precision-recall curve (Fig 4) and it had the highest MCC score of 0.61 showing that this model was the best in correctly classifying the majority of instances of both low and high VVR groups, thus, we used this model to further explore correctly and incorrectly classified samples (see Fig 5) and identify the most important spatial and temporal features that the model's performance was based on.

We evaluated the sensitivity of this model to the occlusion of specific regions of the image (see Fig 6).

Fig 6 shows that the occlusion of the eye and nose regions decreased the performance on the test set drastically (F1 score dropped to 0.27–0.13 around those regions) and that therefore the eye and nose regions are crucial for predicting low and high VVR classes. Conversely, mouth and forehead areas were only moderately affected by occlusion (F1 score dropped to 0.5–0.68).

## VVR regression results

As a regression task, we applied a 2D CNN model with GRU and LSTM on various lengths of image sequences to directly predict VVR scores. The overview of the results is given in Table 2.

The lowest achieved RMSE on the test set was 2.56 using a pre-trained Xception model with GRU on 50 frames sequence or, in other words, 10 seconds of video recording.

## Discussion

In this study, we assessed whether it is possible to predict VVR levels from facial videos using 2D-CNN models with GRU and LSTM, on as short as possible video lengths. The results showed that the best performance in the classification task was achieved using a pre-trained ResNet152 model with LSTM. However, these results were only slightly better than using a pre-trained ResNet152 model with GRU or a pre-trained Xception model with both GRU and

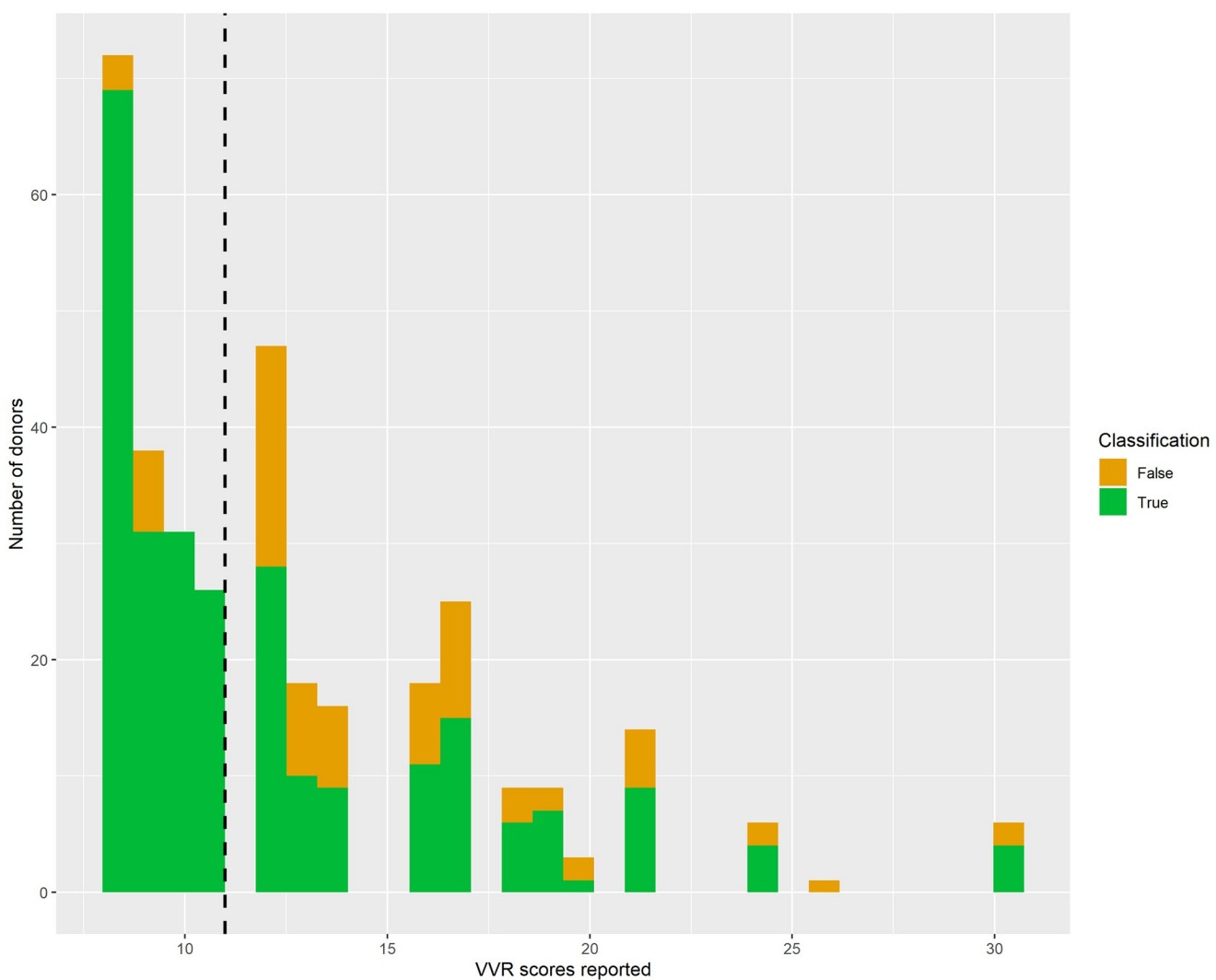

**Fig 5. Performance of ResNet152 with LSTM on test set.** The figure shows correctly and incorrectly classified samples on the test set using a 2D CNN pre-trained model on ResNet152 with LSTM on 25-frame video sequences. The dashed line in separates low (on the left side) and high (on the right side) VVR groups.

LSTM. In addition, the performance in the regression task was higher using GRU than LSTM models. As the performance of GRU is also almost 30% faster [75], using a GRU model would be more beneficial in real-time applications.

For both classification and regression tasks, shorter video lengths resulted in a similar or even slightly better performance. In the classification task, the best-performing model reached an F1 score of 0.74 in the high VVR group using the shortest sequence of 5 seconds (25 frames) of video. As the F1 score is the overall harmonic mean of precision and recall, this measures the ability of the model to both capture high VVR examples (recall) and be accurate with those cases that the model captured (precision). In the regression task, the lowest achieved Root Mean Square Error on the test set was 2.56 using an Xception model on 10 seconds (50 frames) as an input. The RMSE shows how much the predictions made by the model differ from the predicted data. The normalized RMSE of 0.08 (calculated as the RMSE divided by the difference between the maximum and minimum VVR scores) could be interpreted as very low,

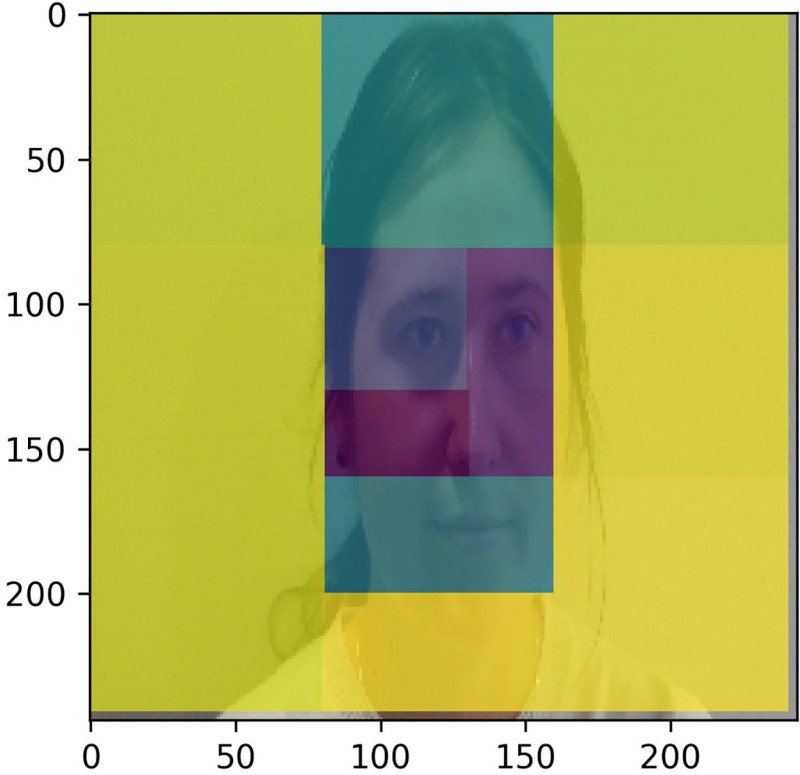

**Fig 6. The illustration of the overall F1 score on the test set after specific regions of the image were occluded using ResNet152 with GRU on 25 frames video sequence.** Note: a darker colour indicates that the performance goes down when occluding that part of the image.

indicating a very high performance of the model. However, a more realistic reflection of the model performance would probably be obtained by dividing the RMSE by the mean VVR score, resulting in a normalized RMSE value of 0.20, which shows moderate performance. Nevertheless, the error rate of 2.56 on a scale ranging from 8 to 40 with a mean value of 11.60 can be considered relatively moderate.

To ensure robustness of our results, we conducted additional investigations with a nested 5-fold cross-validation with inner k = 3 for testing hyperparameters and outer k = 5 for evaluating model performance (see Supplementary material for the results). The results are consistent across tested lengths, folds, and models: the average F1 score across 5 tested folds on classification task using both ResNet152 with GRU and ResNet152 with LSTM was 0.68, and the best F1 score achieved using ResNet152 with GRU was 0.71 and using ResNet152 with

**Table 2. The 2D-CNN performance on test sets using a pre-trained Xception model with GRU and LSTM.**

|  | Video length (N = number of frames) | RMSE on the test set | |
|---|---|---|---|
|  |  | **GRU** | **LSTM** |
| **Pre-trained on Xception** | N = 25 | 2.74 | 2.97 |
|  | **N = 50** | **2.56** | 3.03 |
|  | N = 100 | 2.95 | 2.95 |
|  | N = 150 | 2.90 | 2.93 |
|  | N = 225 | 2.94 | 2.80 |

LSTM was 0.70 (the average results are reported in S1 Table in S1 File and the best performance along with tested hyperparameters is reported in S2 Table in S1 File). This is in line with the presently reported F1 scores in this study which varied from 0.67 to 0.83. In both main and additional analysis we found that pre-trained ResNet152 models performed better than Xception models, however, the performance between GRU and LSTM models was similar. Although we concluded that ResNet152 with LSTM was the best model, we found no statistically significant differences between the ResNet152 with GRU and ResNet152 with LSTM models in the present and additional analysis. As a practical solution for people with needle fear, the main objective should be to correctly capture as many donors who belong to the high VVR group as possible using the shortest duration. The shortest duration (e.g., 25 frames instead of 225 frames) would increase the frequency of providing feedback by the model, which may allow individuals to find the most suitable way of immediately responding to any physical or psychological changes related to their fear. Also, a shorter recording would require the collection of less data, have a more dynamic response in the game, and enable speedier feedback. Having said that, we first assessed which model performed well in identifying the high VVR group. The best balance in classifying both groups was achieved using ResNet152 with LSTM with an MCC score of 0.61. The balance between correctly identifying the low and high VVR groups is preferred. However, given that there are misclassifications in the current model, the preference is to capture as many individuals at risk of experiencing high VVR symptoms even if the model sometimes incorrectly labels those who are *not* at risk. This is important because those who may experience vasovagal reactions, are more likely to develop needle fear and have repeat vasovagal reactions [1,10,19]. Thus, since ResNet152 showed a slightly higher performance in the low VVR group reaching an F1 score of 0.83, the same model showed also high performance in the high VVR group reaching an F1 score of 0.74. Therefore, we concluded that the ResNet152 with an F1 score of 0.74 on 25 frames is preferred for further investigation.

Our results show that the nose and eye regions are the most predictive. This corroborates previous findings from a subset of the same data showing that micro-expressions around the eyes (specifically eyelid raiser and tightener) and eyebrows (specifically, brow lowered) where the most predictive action units of the best-performing machine learning model [34]. Furthermore, in another study using infrared thermal imaging in a virtual blood donation setting with student participants, we also found that donors who experienced high VVR symptoms during the donation tended to show greater velocity of thermal fluctuations around the nose, chin, and forehead areas [32,33,41]. In the future, we could focus on a smaller facial area such as eye and nose regions for extracting facial features. This potentially can provide better model performance, and also use fewer computational resources in the real-world applications in comparison to using the facial features extracted from the entire face.

One of the limitations of our study was a skewed dataset. The majority of people reported low VVR scores. Even though the number of donors who report high VVR symptoms reflects the overall prevalence of VVR symptoms (e.g. [76–78]) the class imbalance may have negatively impacted the performance of the classification models. For example, the model made very few mistakes in identifying donors who belonged to the low VVR class, and the majority of mistakes were in identifying donors who belonged to the high VVR class. Even though this imbalance was addressed by applying a data augmentation technique where we generated new samples for the minority class in the training set, this may not be sufficient. Hence, it would be essential to obtain a more diverse and balanced dataset to evaluate whether model performance could be improved further.

In this study, we used 2D-CNN with LSTM and GRU models and did not explore, for instance, 3D-CNN or hybrid models that could combine both spatial and temporal data

streams simultaneously. Moreover, we could collect psychophysiological data such as heart rate, respiration and blood pressure in order to incorporate multiple streams of information or compare their performances. Finally, as the prevalence rates of fear and VVR are higher in more general samples, it would be ideal if more data could be collected in the future among individuals who are at high risk of experiencing vasovagal reactions e.g. patients undergoing blood draws or immunizations. Some of these limitations will be remedied in the future, as we are currently repeating the virtual rubber arm experiment study. We not only use both thermal and RGB cameras, but also include psychophysiological measurements. This study will also allow us to invite people from the general population, potentially allowing us to collect a more diverse sample.

In conclusion, our results demonstrate that using facial information from video recordings as short sequence as 5 seconds can be used to distinguish high and low VVR responses in blood donors and that this method could be used for predicting VVR responses in a non-intrusive, contactless manner.

## Supporting information

**S1 File.** S1 Table. An average estimate of model performance across 5-fold splits using pre-trained Xception and ResNet152 with GRU and LSTM on various video lengths in classifying low and high (minority class) VVR groups. S2 Table. The 2D-CNN best model performance on the test split classifying low vs high VVR classes using pre-trained Xception and ResNet152 models with GRU and LSTM on various video sequences ranging from 150 to 25 frames. S3 Table. The obtained F1 score across all tested folds using the shortest video duration (N = 25 frames).
(DOCX)

## Acknowledgments

We thank Hasti Memarzadeh for collecting the data, Natalie de Wit and Laura Heij for her support in preprocessing the data, all blood donors who voluntarily participated in our study, and the staff at the participating blood collection centers for their hospitality in hosting this study.

## Author Contributions

**Conceptualization:** Judita Rudokaite, Elisabeth Huis in 't Veld.

**Data curation:** Judita Rudokaite.

**Formal analysis:** Judita Rudokaite.

**Funding acquisition:** Elisabeth Huis in 't Veld.

**Investigation:** Judita Rudokaite.

**Methodology:** Judita Rudokaite, Sharon Ong, Itir Onal Ertugrul, Mart P. Janssen, Elisabeth Huis in 't Veld.

**Project administration:** Elisabeth Huis in 't Veld.

**Resources:** Elisabeth Huis in 't Veld.

**Software:** Judita Rudokaite.

**Supervision:** Sharon Ong, Itir Onal Ertugrul, Mart P. Janssen, Elisabeth Huis in 't Veld.

**Validation:** Sharon Ong, Itir Onal Ertugrul, Mart P. Janssen, Elisabeth Huis in 't Veld.

**Visualization:** Judita Rudokaite.

**Writing – original draft:** Judita Rudokaite.

**Writing – review & editing:** Judita Rudokaite, Sharon Ong, Itir Onal Ertugrul, Mart P. Janssen, Elisabeth Huis in 't Veld.

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
