## [Decision Letter · Decision Letter 0]

19 Feb 2024

PONE-D-23-40599Predicting vasovagal reactions to needles from video data using 2D-CNN with GRU and LSTM.PLOS ONE

Dear Dr. Rudokaite,

Thank you for submitting your manuscript to PLOS ONE. After careful consideration, we feel that it has merit but does not fully meet PLOS ONE’s publication criteria as it currently stands. Therefore, we invite you to submit a revised version of the manuscript that addresses the points raised during the review process.

This paper proposes an ML-based pipeline to study vasovagal reactions to medical procedures involving needles. According to the authors, this is a first-of-a-kind paper in this field, and, in addition, the authors are working on getting a new dataset relevant to this study. The idea is to record short videos of patients and provide them with a questionnaire to note their reactions. The authors then annotate each video (VVR reaction) based on these responses. They use two well-known pre-trained networks to extract the video features and use them further on sequence models for the reaction classification task. 

This is an important research work in medical science using AI, and there are some interesting questions and queries from the learned reviewers. The reviewers raised some strong concerns, both in the context of medical science and deep network-based modelling, and these questions must be addressed in a rebuttal. Furthermore, I read this paper quite thoroughly. Here are my comments; I hope the authors find them helpful in improving their writeup and research.

First, I enjoyed reading the manuscript; it is short and easy to follow and comprehend; the literature review is well done, and the results and downstream analysis sound reasonable. I do have some comments on that; see below. 

The paper is rather organised in a highly complicated way. It is very difficult to know what are various subsections are under each section. This makes reading a bit difficult for a reader. For a journal article, this needs to be done in a manner that reads like a good scientific story. The introduction seems all over the play, although connected to some extent. It can be improved with different subsections in place. 

I am not quite sure whether ethics approval should come directly within the methods section or should be moved to the section right after the conclusion. Please follow the journal guidelines while preparing the manuscript. 

It is also better to provide a dataset summary table that can reduce a lot of text or the texts can be moved to the table's caption. I would like to see a panel figure that visually describes each stage of the data collection methodology or the entire pipeline. 

For datasets of this kind, it is very important that the samples are distributed very rationally into train/validation/test sets. Sometimes it becomes a matter of cherry-picking to show a better performance. To have a better estimate of the performance, it is preferred, in such small datasets, to perform a k-fold cross-validation (k=5 is usually good) and provide an average estimate of the performance. This doesn't overestimate nor underestimate the model. Furthermore, since this is a highly skewed dataset, such an experiment is warranted. The authors might want to do that. Please also ensure that there is no data leakage from the training to the testing set while doing the modelling. Since this is the first study in this area, it is better to show a good modelling strategy that the research community can follow.

As a paper focusing on medical domain applications, one would like to see substantial discussion on those and how previous AI or non-AI-based studies have done those.

Minor and typos:

RMSEprop  RMSPropProofreading is needed to improve the presentation. ==============================

We look forward to receiving your revised manuscript.

Kind regards,

Tirtharaj Dash

Academic Editor

PLOS ONE

Journal Requirements:

This research was funded by the ZonMW Veni project “FAINT” (project reference: 016.186.020) and Stichting Sanquin Bloedvoorziening (grant PPOC19-12/L2409).

Authors Dr. E.M.J. Huis in ‘t Veld and J. Rudokaite are founders of AINAR B.V. The other authors declare no competing interests.

We note that one or more of the authors are employed by a commercial company: AINAR B.V.

“The funder provided support in the form of salaries for authors, but did not have any additional role in the study design, data collection and analysis, decision to publish, or preparation of the manuscript. The specific roles of these authors are articulated in the ‘author contributions’ section.”

Please include both an updated Funding Statement and Competing Interests Statement in your cover letter. We will change the online submission form on your behalf."

4. In the online submission form, you indicated that the dataset collected during the current study is not publicly available due to participants’ privacy, but an anonymized preprocessed dataset can be requested when the FAINT study is completed, by contacting the data manager of the Donor Medicine Research department at Sanquin. For contact details, contact the corresponding author or dr. Elisabeth Huis in ‘t Veld.

5. We note that Figures 2, 3 and 7 includes an image of a participant in the study. 

Additional Editor Comments:

This paper proposes an ML-based pipeline to study vasovagal reactions to medical procedures involving needles. According to the authors, this is a first-of-a-kind paper in this field, and, in addition, the authors are working on getting a new dataset relevant to this study. The idea is to record short videos of patients and provide them with a questionnaire to note their reactions. The authors then annotate each video (VVR reaction) based on these responses. They use two well-known pre-trained networks to extract the video features and use them further on sequence models for the reaction classification task.

This is an important research work in medical science using AI, and there are some interesting questions and queries from the learned reviewers. The reviewers raised some strong concerns, both in the context of medical science and deep network-based modelling, and these questions must be addressed in a rebuttal. Furthermore, I read this paper quite thoroughly. Here are my comments; I hope the authors find them helpful in improving their writeup and research.

First, I enjoyed reading the manuscript; it is short and easy to follow and comprehend; the literature review is well done, and the results and downstream analysis sound reasonable. I do have some comments on that; see below.

The paper is rather organised in a highly complicated way. It is very difficult to know what are various subsections are under each section. This makes reading a bit difficult for a reader. For a journal article, this needs to be done in a manner that reads like a good scientific story. The introduction seems all over the play, although connected to some extent. It can be improved with different subsections in place.

I am not quite sure whether ethics approval should come directly within the methods section or should be moved to the section right after the conclusion. Please follow the journal guidelines while preparing the manuscript.

It is also better to provide a dataset summary table that can reduce a lot of text or the texts can be moved to the table's caption. I would like to see a panel figure that visually describes each stage of the data collection methodology or the entire pipeline.

For datasets of this kind, it is very important that the samples are distributed very rationally into train/validation/test sets. Sometimes it becomes a matter of cherry-picking to show a better performance. To have a better estimate of the performance, it is preferred, in such small datasets, to perform a k-fold cross-validation (k=5 is usually good) and provide an average estimate of the performance. This doesn't overestimate nor underestimate the model. Furthermore, since this is a highly skewed dataset, such an experiment is warranted. The authors might want to do that. Please also ensure that there is no data leakage from the training to the testing set while doing the modelling. Since this is the first study in this area, it is better to show a good modelling strategy that the research community can follow.

As a paper focusing on medical domain applications, one would like to see substantial discussion on those and how previous AI or non-AI-based studies have done those.

Minor and typos:

* RMSEprop  RMSProp

* Proofreading is needed to improve the presentation.

Reviewers' comments:

Reviewer's Responses to Questions

**Comments to the Author**

1. Is the manuscript technically sound, and do the data support the conclusions?

Reviewer #1: Partly

Reviewer #2: Partly

Reviewer #3: Partly

2. Has the statistical analysis been performed appropriately and rigorously? 

Reviewer #1: Yes

Reviewer #2: No

Reviewer #3: Yes

3. Have the authors made all data underlying the findings in their manuscript fully available?

Reviewer #1: Yes

Reviewer #2: No

Reviewer #3: Yes

4. Is the manuscript presented in an intelligible fashion and written in standard English?

Reviewer #1: No

Reviewer #2: No

Reviewer #3: Yes

5. Review Comments to the Author

Reviewer #1: 1. How does the study align with existing theories on the interplay between psychological states, physiological responses, and the predictability of vasovagal reactions during needle-related procedures?

2.Considering the prevalence of severe vasovagal reactions, how does the study address the challenge of training machine learning models with limited data?

3.How does the proposed serious game intervention aim to utilize facial information obtained through video recordings, and what advantages does visible light imaging offer over infrared thermal imaging?

4.What informed the choice of pre-trained models such as Xception and ResNet152 for feature extraction, and how do they contribute to the classification and regression tasks in predicting VVR levels?

5.Can you elaborate on the specific physiological changes, such as increased heart rate and changes in facial pallor, associated with vasovagal reactions?

Reviewer #2: In this paper, the authors have used 2D-CNN with LSTM and GRU to predict continuous VVR scores and to classify

discrete (low and high) VVR values obtained during the blood donation. Although the study has practical significance, the result discussion section and the English could be further improved. This paper may be considered if the following comments are carefully addressed.

1. The text is written a bit carelessly. Proofread it.

2. The employed LSTM and GRU model parameters should be provided.

3. Tune the hyperparameters of used employed LSTM and GRU models for improved performance.

4. Although stochastic, 2D-CNN, LSTM and GRU models are employed in this paper, no statistical analysis on the obtained results are carried out.

5. It is suggested to repeat the simulations and apply non-parametric statistical tests on the obtained results to draw reliable conclusions.

6. The figure quality of figure 1,2, 5 need improvement.

Reviewer #3: In the study, Rudokaite et al have investigated the prediction of vasovagal reactions (VVR) during needle-related procedures by analyzing facial information from video sequences obtained from 287 blood donors. Further, utilizing 2D-CNN with LSTM and GRU, authors achieved a notable F1 score of 0.74 for high VVR classification using a pre-trained ResNet152 model with LSTM on 25 frames, and demonstrated the feasibility of predicting VVR responses during blood donation, suggesting potential applications for preventive interventions. Below are few comments for consideration

1- The authors have provided an elaborate introduction. It could be made more concise and precise to offer sufficient context for the study.

2- Authors should also discuss the age range of the donors and if at all there was any difference.

3- Authors should elaborate on the high VVR class, and what implications do the model's performance metrics have for identifying individuals at risk of experiencing VVR symptoms?

4- A discussion on the practical implications of the study's findings is needed . How could the identified predictive features in the nose and eye regions be practically applied? What are the potential real-world applications and benefits of using this method for predicting VVR responses?

5-A brief mention of potential future work or directions for further research. Are there specific areas or aspects that could be explored in subsequent studies?

6. PLOS authors have the option to publish the peer review history of their article (what does this mean?). If published, this will include your full peer review and any attached files.

Reviewer #1: **Yes: **suresh a

Reviewer #2: **Yes: **Sibarama Panigrahi

Reviewer #3: No

---

## [Author Response · Author response to Decision Letter 0]

2 Aug 2024

Dear editor and reviewers,

We thank both reviewers and the editor for their careful consideration of the manuscript, and for their knowledgeable and constructive comments. We’ve addressed all your suggestions for improvements. You can find a summary of our improvements in relationship to each comment below (marked in red).

Editor comments.

1. The paper is rather organised in a highly complicated way. It is very difficult to know what are various subsections are under each section. This makes reading a bit difficult for a reader. For a journal article, this needs to be done in a manner that reads like a good scientific story. The introduction seems all over the play, although connected to some extent. It can be improved with different subsections in place. 

We have adjusted the names of the subsections and adjusted the introduction to improve the readability. 

2. I am not quite sure whether ethics approval should come directly within the methods section or should be moved to the section right after the conclusion. Please follow the journal guidelines while preparing the manuscript. 

Thank you for raising this question. We verified that the ethics approval should be included in the methods section as the journal guidelines state that “methods sections of papers on research using human subjects or samples must include ethics statements” (https://journals.plos.org/plosone/s/human-subjects-research). However, in the revised manuscript we include the ethics statement after the description of the participants.

3. It is also better to provide a dataset summary table that can reduce a lot of text or the texts can be moved to the table's caption. I would like to see a panel figure that visually describes each stage of the data collection methodology or the entire pipeline. 

We have provided the entire pipeline from data collection stage to data analysis (see Figure 1).

Fig. 1. An overview of the data collection and used methodology for data analysis. Stage 1: Data collection shows an overview of the blood donation testing procedure and stages. At each of the seven stages, donors reported their VVR levels, and a video recording was made. At each stage, the video recording lasted 1 to 2 minutes. From stage 4 until stage 6, the recording was continuous, lasting between 5 and 27 minutes. *The procedure is slightly different per BCC. At two locations, donors were brought directly to the donation chair after the physician check and hence no recording at stage 3 was made. If donors had to return to the waiting area after the physician check, they would be asked for an additional self-reported VVR score. An additional recording of 1 to 2 minutes would be made. Stage 2: Data preprocessing shows an overview of the video preprocessing steps. At each step, the earliest frames were removed. i.e. the frames closer to self-report VVR measures are kept for the analysis. In addition, the number of frames was reduced from 30 FPS to 5 FPS. Stage 3: Data analysis shows an overview of model training and evaluation steps.

4. For datasets of this kind, it is very important that the samples are distributed very rationally into train/validation/test sets. Sometimes it becomes a matter of cherry-picking to show a better performance. To have a better estimate of the performance, it is preferred, in such small datasets, to perform a k-fold cross-validation (k=5 is usually good) and provide an average estimate of the performance. This doesn't overestimate nor underestimate the model. Furthermore, since this is a highly skewed dataset, such an experiment is warranted. The authors might want to do that. Please also ensure that there is no data leakage from the training to the testing set while doing the modelling. Since this is the first study in this area, it is better to show a good modelling strategy that the research community can follow.

We agree with your raised concerns. We have followed your advice and re-trained our models using stratified 5-k fold validation technique to ensure that the model performance is not overly optimistic. 

“We used a 5-fold stratified validation technique on the training set, in which the dataset is divided into 5 subsets, and the model is trained and tested 5 times. In each iteration, one subset is used for testing, and the remaining 4 subsets are used for training. Stratified cross-validation was selected to ensure that each fold contains a representive sample of all classes, which is particularly important due to dataset imbalance to reduce the risk of biased performance. 

We have also separated our data “based on subject identification number to ensure that the same participants would not appear in both training and testing sets.”

In conclusion, our results were consistent across all tested folds and the best performance was in line with our previously reported findings. We hope that this addresses your concerns. 

We included the results in the Supplementary Material. We reported the average F1 score with SD as well as the highest precision, recall, F1, PR-AUC, and MCC scores obtained. Finally, we reported the actual scores of F1 across all tested folds on the shortest video duration for all four models and compared the model performance using the non-parametric Friedman test. 

S1 Appendix. To ensure model robustness and reduce the risk of biased performance, we also completed an additional analysis where we used a nested 5-fold stratified validation technique, in which the dataset is divided into 5 subsets, and the model is trained and tested 5 times. In each iteration, one subset is used for testing, and the remaining 4 subsets are used for training. Stratified cross-validation was selected to ensure that each fold contains a representative sample of all classes, which is particularly important due to dataset imbalance. In addition, the data was split based on subject identification number to ensure that the same participants would not appear in both training and testing sets. A nested cross-validation was selected to optimize the hyperparameters and avoid overfitting. We used k=5 for the outer loop (model evaluation) and a k=3 for the inner loop (hyperparameter testing). In S1 Table we report the average F1 score with standard deviation obtained on the tested split for all tested folds, and in S2 Table we report the results obtained on the best performing model along with used hyperparameters.

S1 Table. An average estimate of model performance across 5-fold splits using pre-trained Xception and ResNet152 with GRU and LSTM on various video lengths in classifying low and high (minority class) VVR groups.

Model Number of frames Average F1 Score across Tested Splits Standard Deviation of F1 Scores across Tested Splits

Pre-trained Xception model with GRU N = 150 0.66 0.02

 N = 100 0.66 0.04

 N = 50 0.51 0.25

 N = 25 0.63 0.02

Pre-trained Xception model with LSTM N = 150 0.66 0.02

 N = 100 0.65 0.01

 N = 50 0.66 0.02

 N = 25 0.62 0.04

Pre-trained ResNet152 model with GRU N = 150 0.68 0.01

 N = 100 0.68 0.02

 N = 50 0.66 0.03

 N = 25 0.67 0.03

Pre-trained ResNet152 model with LSTM N = 150 0.68 0.02

 N = 100 0.68 0.02

 N = 50 0.67 0.02

 N = 25 0.67 0.02

S2 Table. The 2D-CNN best model performance on the test split classifying low vs high VVR classes using pre-trained Xception and ResNet152 models with GRU and LSTM on various video sequences ranging from 150 to 25 frames.

Model Number of frames Precision Recall F1 AUC-PR MCC Hyperparameters used

Pre-trained Xception model with GRU N = 150 0.75 0.63 0.68 0.65 0.43 'batch_size': 64, 'dropout_rate': 0.1, 'epochs': 100, 'learning_rate': 0.001

 N = 100 0.74 0.66 0.70 0.65 0.46 'batch_size': 64, 'dropout_rate': 0.1, 'epochs': 100, 'learning_rate': 0.001

 N = 50 0.63 0.67 0.65 0.57 0.36 'batch_size': 32, 'dropout_rate': 0.1, 'epochs': 200, 'learning_rate': 0.0001

 N = 25 0.71 0.61 0.66 0.61 0.40 'batch_size': 32, 'dropout_rate': 0.5, 'epochs': 200, 'learning_rate': 0.0001

Pre-trained Xception model with LSTM N = 150 0.76 0.63 0.68 0.65 0.45 'batch_size': 64, 'dropout_rate': 0.5, 'epochs': 100, 'learning_rate': 0.0001

 N = 100 0.63 0.71 0.67 0.57 0.38 'batch_size': 64, 'dropout_rate': 0.1, 'epochs': 100, 'learning_rate': 0.0001

 N = 50 0.68 0.71 0.69 0.60 0.43 'batch_size': 64, 'dropout_rate': 0.1, 'epochs': 200, 'learning_rate': 0.0001

 N = 25 0.73 0.61 0.67 0.64 0.40 'batch_size': 32, 'dropout_rate': 0.1, 'epochs': 100, 'learning_rate': 0.0001

Pre-trained ResNet152 model with GRU N = 150 0.69 0.68 0.69 0.61 0.44 'batch_size': 64, 'dropout_rate': 0.5, 'epochs': 200, 'learning_rate': 0.001

 N = 100 0.75 0.68 0.71 0.66 0.48 'batch_size': 64, 'dropout_rate': 0.1, 'epochs': 100, 'learning_rate': 0.001

 N = 50 0.78 0.64 0.70 0.67 0.49 'batch_size': 32, 'dropout_rate': 0.5, 'epochs': 100, 'learning_rate': 0.001

 N = 25 0.65 0.79 0.71 0.61 0.41 'batch_size': 32, 'dropout_rate': 0.5, 'epochs': 200, 'learning_rate': 0.001

Pre-trained ResNet152 model with LSTM N = 150 0.73 0.81 0.71 0.60 0.40 'batch_size': 64, 'dropout_rate': 0.5, 'epochs': 100, 'learning_rate': 0.001

 N = 100 0.67 0.71 0.69 0.61 0.38 'batch_size': 64, 'dropout_rate': 0.5, 'epochs': 100, 'learning_rate': 0.001

 N = 50 0.75 0.63 0.69 0.66 0.44 'batch_size': 64, 'dropout_rate': 0.5, 'epochs': 200, 'learning_rate': 0.001

 N = 25 0.77 0.63 0.70 0.67 0.47 'batch_size': 64, 'dropout_rate': 0.1, 'epochs': 100, 'learning_rate': 0.001

5. As a paper focusing on medical domain applications, one would like to see substantial discussion on those and how previous AI or non-AI-based studies have done those.

 In our introduction, we included a section on current interventions used for preventing vasovagal reactions and what are the limitations. For example: 

“Applied muscle tension consists of repeated contractions of muscles in the legs and/or abdomen in order to increase blood pressure [11, 13]. Even though research shows that these techniques could work for a subset of donors [13], a meta-analysis suggested that these techniques are insufficient for the majority of donors [11] and do not reduce the rate of syncopic reactions [13]. 

Other preventive strategies provided by healthcare professionals range from providing extra information, social support or distractions or even administering calming medication such as low doses of benzodiazepines that allow the patients to reduce their anxiety levels (Gebhardt et al., 2018). Although benzodiazepines reduce the number of vasovagal reactions by addressing underlying fear and anxiety, some side effects might make them less favorable as a prevention strategy, especially, as blood draws or immunizations are quick procedures. All previously applied preventive techniques might be effective, however, they are costly in terms of extra time required by the staff. Hence, there is an enormous need for new prevention methods.”

In addition, we included some examples of other papers that used facial video information to estimate the likelihood of experiencing pain, depression, anxiety and other psychological and physiological measures.

“Video recordings of the face contain many types of useful information. For example, it contains information about facial expressions which, even when they are very subtle, enable the detection of anxiety, stress, fear, pain, and vasovagal reactions [33-38]. In addition, the face contains information of head movements, eye-gaze direction, changes in facial colors such as paleness, etc. With the recent developments in the field of deep learning and in particular automatic face analysis [39-41], it has been shown possible to predict not only mental health conditions such as depression, anxiety, or obsessive-compulsive disorder [42], but also physical symptoms such as pain [43-45]. These models can potentially be used as valuable tools for clinical diagnosis and for monitoring and altering physical responses in real time [24]. Specifically, we found a significant association between vasovagal reactions and changes in facial temperature [25] as well as facial micro-expressions [33]. Both changes in thermal fluctuations and facial action units recorded prior to blood donation showed promising results in predicting vasovagal reactions that occur during or after blood donation [24, 25, 33].”

6. Minor and typos: 

RMSEprop  RMSProp. Proofreading is needed to improve the presentation.

We have proofread our text and fixed the typos. 

Reviewer #1

Dear Reviewer, 

First of all, we would like to thank you for your valuable insights that allowed us to improve the manuscript. In what follows, we address each comment and demonstrate the changes that were made to the manuscript.

1. How does the study align with existing theories on the interplay between psychological states, physiological responses, and the predictability of vasovagal reactions during needle-related procedures?

Previous studies show that anticipatory anxiety and stress is associated with increased physiological changes such as increased heart rate or cortisol levels that occur at the early stage of blood donation or any other needle-related procedure and can serve as early markers of stress. It is likely that these early subtle psychological and physiological changes could be captured using psychophysiological techniques such as infrared-thermal cameras and, as shown in this experiment, using a standard RGB camera and ML techniques. We included the examples of previous findings in our introduction to elaborate more on the relationship between psychological, physiological changes and vasovagal reactions:

“Research shows that anticipatory fear, anxiety and stress and a history of previous vasovagal reactions are one of the most important risk factors for experiencing vasovagal reactions [16, 17, 10, 18, 19]. This is corroborated by studies from Hoogerwerf et al. (2018; 2017) [20, 21] who assessed psychological, hormonal and psychophysiological stress markers in donors throughout a blood donation and found that the objective stress markers already occur at a very early stage in anticipation of the needle insertion, at which time they peak. For example, the levels of systolic blood pressure and cortisol levels increased towards needle insertion and then decreased after the blood donation [20, 21]. In addition, higher systolic blood pressure and pulse rate were found in women and first-time donors, who are at the higher risk of experiencing VVR [14, 15].”

“In one of our previous studies, we mimicked a blood donation using an experimental 'virtual' rubber arm illusion, imaging the participant with an infrared thermal imaging camera. This experiment showed that changes in facial temperature could serve as early indicators for vasovagal reactions [24, 25]. Specifically, facial temperature fluctuations in the area under the nose, chin and forehead are associated with increased risk of experiencing VVR [25]. “

2.Considering the prevalence of severe vasovagal reactions, how does the study address the challenge of training machine learning models with limited data?

This is indeed a challenge to train machine learning models with limited data of (specifically high VVR) examples. However, we tried a few approaches to improve our model performance. First of all, as we emphasized in our introduction, we used transfer learning to overcome the limited number of samples.

“To train a model from scratch, a large amount of training data of donors with both high and low VVR scores is required, which is difficult to obtain. To overcome this limitation, transfer learning will be applied. Transfer learning is one of the machine learning techniques where a model, initially trained on one task, is used for a different task with some additional tuning.”

Second, to increase the generalization and robustness of the trained models by exposing them to a wider range of variations in the input data, we applied various transformations to the existing video data samples in the training set. We explain this in the section “Model training, validation, and evaluation”:

“Due to class imbalance and limited number of high VVR samples, we applied video data augmentation on the high VVR cases in the training data. Data augmentation is a technique used in

---

## [Decision Letter · Decision Letter 1]

15 Oct 2024

PONE-D-23-40599R1Predicting vasovagal reactions to needles from video data using 2D-CNN with GRU and LSTM.PLOS ONE

Dear Dr. Rudokaite,

Thank you for submitting your manuscript to PLOS ONE. After careful consideration, we feel that it has merit but does not fully meet PLOS ONE’s publication criteria as it currently stands. Therefore, we invite you to submit a revised version of the manuscript that addresses the points raised during the review process.

Dear Authors,

I am happy to read your revised manuscript. I think you have addressed the concerns raised by the esteemed reviewers and me rather very well. I am satisfied with the revision and your response(s) to each of the reviewers' questions. However, after consulting with a domain-expert (new reviewer) and their comments, I am able to propose an acceptance only after you address their concerns, which I think are minor and clarification type. I believe those new comments will certainly improve the manuscript further. I look forward to receiving your revision soon.

Thank you.

We look forward to receiving your revised manuscript.

Kind regards,

Tirtharaj Dash

Academic Editor

PLOS ONE

Journal Requirements:

Additional Editor Comments:

Dear Authors,

I am happy to read your revised manuscript. I think you have addressed the concerns raised by the esteemed reviewers and me rather very well. I am satisfied with the revision and your response(s) to each of the reviewers' questions. However, after consulting with a domain-expert (new reviewer) and their comments, I am able to propose an acceptance only after you address their concerns, which I think are minor and clarification type. I believe those new comments will certainly improve the manuscript further. I look forward to receiving your revision soon.

Thank you.

Reviewers' comments:

Reviewer's Responses to Questions

**Comments to the Author**

1. If the authors have adequately addressed your comments raised in a previous round of review and you feel that this manuscript is now acceptable for publication, you may indicate that here to bypass the “Comments to the Author” section, enter your conflict of interest statement in the “Confidential to Editor” section, and submit your "Accept" recommendation.

Reviewer #4: (No Response)

2. Is the manuscript technically sound, and do the data support the conclusions?

Reviewer #4: Yes

3. Has the statistical analysis been performed appropriately and rigorously? 

Reviewer #4: Yes

4. Have the authors made all data underlying the findings in their manuscript fully available?

Reviewer #4: Yes

5. Is the manuscript presented in an intelligible fashion and written in standard English?

Reviewer #4: Yes

6. Review Comments to the Author

Reviewer #4: This study investigates the prediction of vasovagal reactions (VVR) during blood donation using facial information captured in video sequences. The study demonstrates the potential to predict VVR via facial features to support possible future interventions to prevent adverse reactions. However, there are some possible limitations to consider:

1. In the introduction part - first paragraph, the authors mentioned “these reactions can result in physiological changes such as increased heart rate…”. Then authors mentioned parasympathetic activity such as drops in heart rate or blood pressure. It might be better to bring out only parasympathetic activity directly to avoid confusion.

2. How to define VVR in case group? Are the vasovagal reactions reported by patients themselves or diagnosed by doctors or trained medical staff?

3. How does this quick and short-time identification method prevent VVR? Why is it important? Authors mentioned several potential ways to prevent VVR from happening, but how can identifying VVR prevent VVR? It would be good if authors could describe more on the possible application on medical perspectives.

7. PLOS authors have the option to publish the peer review history of their article (what does this mean?). If published, this will include your full peer review and any attached files.

Reviewer #4: No

---

## [Author Response · Author response to Decision Letter 1]

23 Oct 2024

Dear Editor and Reviewers,

We sincerely thank the new reviewer and the editor for their careful evaluation of our manuscript. We have addressed all of your suggestions for improvement, and a summary of our responses and modifications in relation to each comment is provided below.

1. In the introduction part - first paragraph, the authors mentioned “these reactions can result in physiological changes such as increased heart rate…”. Then authors mentioned parasympathetic activity such as drops in heart rate or blood pressure. It might be better to bring out only parasympathetic activity directly to avoid confusion.

Thank you for pointing this out. Indeed this can be misleading, therefore, we adjusted the text to better explain how the vasovagal reactions occur: 

“These so-called vasovagal reactions (VVR) can manifest as nausea, dizziness, heart palpitations, hyperventilation, or even fainting with a temporary loss of consciousness. They occur when the body experiences an exaggerated response to a trigger such as blood draws or injections during the medical procedure. The VVR reactions are primarily mediated by the autonomic nervous system, which controls involuntary functions like heart rate and blood pressure. Initially, the sympathetic branch of the autonomic nervous system activates in response to the stressful or fearful situation, as part of the body's "fight or flight" mechanism, leading to an increase in heart rate [2-3], sweating, nausea, and pupil dilation [4, 5, 6]. However, in some individuals, the parasympathetic nervous system, which serves as the "rest and digest" counterpart, overcompensates for this initial response, working to slow down the heart rate and lower blood pressure in an attempt to restore balance. When this parasympathetic response becomes too strong, it can cause a rapid drop in heart rate and blood pressure [3], reducing blood flow to the brain. This decrease in blood and oxygen supply can result in dizziness, nausea, and ultimately fainting (syncope) [7].”

2. How to define VVR in case group? Are the vasovagal reactions reported by patients themselves or diagnosed by doctors or trained medical staff?

The vasovagal reactions were self-reported by patients. We now specified this in the section called Procedure:

“Specifically, at each stage donors were recorded using a regular video camera and had to verbally self-report their VVR score.”

And under section Materials and Methods. Vasovagal reactions:

“participants were asked to verbally rate 8 questions regarding experienced physiological reactions”

3. How does this quick and short-time identification method prevent VVR? Why is it important? Authors mentioned several potential ways to prevent VVR from happening, but how can identifying VVR prevent VVR? It would be good if authors could describe more on the possible application on medical perspectives.

Thank you for your question. It seems this was not well explained in the text. We modified the “Applying deep learning methods for automated video analysis for a biofeedback-based serious game intervention” section in the Introduction, to better explain on how the game would use information of the risk of experiencing VVRs:

“In order to address the lack of interventions that address the anticipatory risk factors for VVR in practice we developed a solution for people with needle fear that is able to not only identify covert symptoms of emotional and physical reactions at a very early stage, but also to immediately give them a tool which can help them to prevent the escalation into a vasovagal event. To achieve that, we aimed to implement the best performing model in a serious game for smartphones, which through the facial video input from the front-facing camera, controls a biofeedback mechanism which will help the player get control over their impeding VVR in an early stage.

<...>

However, assessing the risk of VVR is only one part of the solution. Even better would be if the patient can use this information to prevent the VVR from happening. This can be achieved through biofeedback. <....>

By seeing or hearing these physiological signals in real time, patients can experiment with strategies to manage their stress or anxiety and bring their physiological state back into balance. <...> In these games, the stimuli adjust based on the player’s bodily responses, allowing the individual to practice managing their physiological reactions in a controlled and engaging environment.

In the case of needle fear, our serious game solution (called AINAR, Artificial Intelligence for Needle Induced Fainting) continuously assesses the likelihood of experiencing a vasovagal reaction (VVR) through the model, which gets its input from the front-facing camera, which is then reflected in the weather, which can be sunny, rainy, or snowy. The player's task is to keep the weather nice and sunny. If it starts to rain, the player can experiment with different relaxation techniques or cognitive strategies to transition from a state of fear or stress to calmness, thus learning to control their body's automatic responses. The aim is to provide the feedback as often as possible with little delay, therefore, we aim to investigate what is the shortest length of video that is required for acceptable VVR prediction. This approach allows for real-time learning and adaptation, making biofeedback a powerful tool for overcoming anxiety and stress-related conditions.”

---

## [Decision Letter · Decision Letter 2]

5 Nov 2024

Predicting vasovagal reactions to needles from video data using 2D-CNN with GRU and LSTM.

PONE-D-23-40599R2

Dear Dr. Rudokaite,

We’re pleased to inform you that your manuscript has been judged scientifically suitable for publication and will be formally accepted for publication once it meets all outstanding technical requirements.

Kind regards,

Mohammad Amin Fraiwan

Academic Editor

PLOS ONE

Additional Editor Comments (optional):

Reviewers' comments:

Reviewer's Responses to Questions

**Comments to the Author**

1. If the authors have adequately addressed your comments raised in a previous round of review and you feel that this manuscript is now acceptable for publication, you may indicate that here to bypass the “Comments to the Author” section, enter your conflict of interest statement in the “Confidential to Editor” section, and submit your "Accept" recommendation.

Reviewer #4: All comments have been addressed

2. Is the manuscript technically sound, and do the data support the conclusions?

Reviewer #4: Yes

3. Has the statistical analysis been performed appropriately and rigorously? 

Reviewer #4: Yes

4. Have the authors made all data underlying the findings in their manuscript fully available?

Reviewer #4: Yes

5. Is the manuscript presented in an intelligible fashion and written in standard English?

Reviewer #4: Yes

6. Review Comments to the Author

Reviewer #4: (No Response)

7. PLOS authors have the option to publish the peer review history of their article (what does this mean?). If published, this will include your full peer review and any attached files.

Reviewer #4: No

---

## [Editor Report · Acceptance letter]

8 Nov 2024

PONE-D-23-40599R2 

PLOS ONE

Dear Dr. Rudokaite, 

I'm pleased to inform you that your manuscript has been deemed suitable for publication in PLOS ONE. Congratulations! Your manuscript is now being handed over to our production team.

Kind regards, 

on behalf of

Dr. Mohammad Amin Fraiwan 

Academic Editor

PLOS ONE